# Study of the Counter Cation Effects on the Supramolecular Structure and Electronic Properties of a Dianionic Oxamate-Based {Ni^II^_2_} Helicate

**DOI:** 10.3390/molecules28052086

**Published:** 2023-02-23

**Authors:** Cintia A. Simosono, Rafaela M. R. da Silva, Nathália R. De Campos, Marye Agnes R. Silva, Antônio C. Doriguetto, Leonã S. Flores, Charlane C. Correa, Tatiana R. G. Simões, Ana Karoline S. M. Valdo, Felipe T. Martins, Flávio Garcia, Guilherme P. Guedes, Breno R. L. Galvão, Juliana Cancino-Bernardi, Ricardo D. dos Reis, Humberto O. Stumpf, Danielle D. Justino, Paulo F. R. Ortega, Walace D. do Pim, Miguel Julve, Maria Vanda Marinho

**Affiliations:** 1Instituto de Química, Universidade Federal de Alfenas, Campus Santa Clara, Alfenas 37133-840, MG, Brazil; 2Departamento de Química, Universidade Federal de Juiz de Fora, Campus Martelos, Juiz de Jora 36036-900, MG, Brazil; 3Centro Politécnico, Departamento de Química, Universidade Federal do Paraná, Curitiba 81530-900, PR, Brazil; 4Instituto de Física, Universidade Federal de Goiás, Campus Samambaia, Goiânia 74690-900, GO, Brazil; 5Centro Brasileiro de Pesquisas Físicas (CBPF), Rio de Janeiro 22290-180, RJ, Brazil; 6Instituto de Química, Universidade Federal Fluminense, Niterói 24020-141, RJ, Brazil; 7Centro Federal de Educação Tecnológica de Minas Gerais, Departamento de Química, Belo Horizonte 30421-169, MG, Brazil; 8Department of Chemistry, Faculty of Philosophy Sciences and Letters of Ribeirao Preto, University of Sao Paulo, Ribeirão Preto 14040-901, SP, Brazil; 9Brazilian Synchrotron Light Laboratory (LNLS), Brazilian Center for Research in Energy and Materials (CNPEM), Campinas 13083-970, SP, Brazil; 10Departamento de Química—ICEx, Universidade Federal de Minas Gerais, Campus Pampulha, Belo Horizonte 31270-901, MG, Brazil; 11Departament de Química Inorgànica-Instituto de Ciencia Molecular (ICMol), Universitat de València, C/Catedrático José Beltrán 2, Paterna, 46980 València, Spain

**Keywords:** {Ni^II^_2_} helicate, oxamate, supramolecular chemistry, counter cations, electronic properties, redox activity, XANES spectroscopy, theoretical calculations

## Abstract

Herein, we describe the synthesis, crystal structure, and electronic properties of {[K_2_(dmso)(H_2_O)_5_][Ni_2_(H_2_mpba)_3_]·dmso·2H_2_O}_n_ (**1**) and [Ni(H_2_O)_6_][Ni_2_(H_2_mpba)_3_]·3CH_3_OH·4H_2_O (**2**) [dmso = dimethyl sulfoxide; CH_3_OH = methanol; and H_4_mpba = 1,3-phenylenebis(oxamic acid)] bearing the [Ni_2_(H_2_mpba)_3_]^2−^ helicate, hereafter referred to as {Ni^II^_2_}. SHAPE software calculations indicate that the coordination geometry of all the Ni^II^ atoms in 1 and 2 is a distorted octahedron (O_h_) whereas the coordination environments for K1 and K2 atoms in 1 are Snub disphenoid J84 (D_2d_) and distorted octahedron (O_h_), respectively. The {Ni^II^_2_} helicate in **1** is connected by K^+^ counter cations yielding a 2D coordination network with **sql** topology. In contrast to **1**, the electroneutrality of the triple-stranded [Ni_2_(H_2_mpba)_3_] ^2−^ dinuclear motif in **2** is achieved by a [Ni(H_2_O)_6_]^2+^ complex cation, where the three neighboring {Ni^II^_2_} units interact in a supramolecular fashion through four *R_2_^2^*(10) homosynthons yielding a 2D array. Voltammetric measurements reveal that both compounds are redox active (with the Ni^II^/Ni^I^ pair being mediated by OH^–^ ions) but with differences in formal potentials that reflect changes in the energy levels of molecular orbitals. The Ni^II^ ions from the helicate and the counter-ion (complex cation) in **2** can be reversibly reduced, resulting in the highest faradaic current intensities. The redox reactions in **1** also occur in an alkaline medium but at higher formal potentials. The connection of the helicate with the K^+^ counter cation has an impact on the energy levels of the molecular orbitals; this experimental behavior was further supported by X-ray absorption near-edge spectroscopy (XANES) experiments and computational calculations.

## 1. Introduction

The fine-tuning in the design of functional ionic metal-organic materials by a suitable selection of counterions is an exciting pathway to explore while developing new platforms for heterogeneous catalysis [1], gas adsorption [2,3], and electron conductivity [4], to mention just a few. This control can be made using post-synthetic protocols to exchange free guest ions within frameworks, or while controlling the assembling of target materials by choosing the synthetic parameters [5,6,7].

Oxamate-based compounds represent an impressive number of multifunctional molecular materials described in the literature, and a number of publications has shown that the control of several synthetic parameters including metal/linker ratio, temperature, synthetic method, and solvent choice highly contributes to the isolation of materials with varied topologies and properties [8,9,10,11,12,13,14,15]. In the supramolecular coordination materials involving polyoxamate-type ligands, the case of the H_x_mpba^(4-x)-^ family [H_4_mpba = 1,3-phenylenebis(oxamic acid)] has been object of much attention since the built meso-helicates are model systems for the study of self-assembling processes [16,17,18,19]. When using these ligands, their preorganization around the metal centers can lead to the self-assembly of either helicates or mesocates [10,20,21]. The predictable nature of the metal-ligand coordination sphere and the variety, flexibility, and easy functionalization of organic ligands are important factors to achieve new functional architectures through supramolecular chemistry and crystal engineering. This has been shown by the recent development of thousands of fascinating materials called porous coordination polymers (PCPs) or metal-organic frameworks (MOFs) [22,23,24].

The H_x_mpba^(4-x)–^ ligand can be found in metal-organic structures either in the partially or fully deprotonated forms adopting mainly a tetradentate coordination mode towards the metal centers yielding polynuclear complexes. Classical examples are the heterochiral honeycomb-like frameworks with a triple-stranded dinuclear core (see Appendix A). It has been exploited since the first report by Pardo *et al.* [25], and henceforth, different facets of di-deprotonated (H_2_mpba^2−^) and fully deprotonated (mpba^4−^) compounds have been uncovered (Appendix A) [16,17,18,20,21,26,27,28,29,30,31,32,33]. From this family of compounds, those containing the [Ni_2_(H_2_mpba)_3_]^2−^ motif have been scarcely explored, the first reported example being the complex of formula Na_8_[Ni_2_(mpba)_3_]∙10H_2_O [18]. To the best of our knowledge and in a journey through the last three decades, besides the systems described in this contribution, there are only four other examples of isolated Ni^II^-based compounds containing the di-deprotonated (H_2_mpba^2−^) or fully deprotonated (mpba^4−^) forms of the 1,3-phenylenebis(oxamic) acid (Appendix A).

Herein we describe the role played by different cations {K^+^ and [Ni(H_2_O)_6_]^2+^} on the topology of supramolecular materials based on the triple-stranded [Ni_2_(H_2_mpba)_3_]^2−^ dinuclear motif, hereafter referred to as {Ni^II^_2_} (see Figure 1).

We show in this study that the choice of solvents used in the preparation of the supramolecular materials based on this {Ni^II^_2_} unit seems to dictate the type of cation to be incorporated into the crystal structures. We found that the use of a H_2_O/MeOH/dmso—(1:1:1 *v*/*v*/*v*) solvent mixture yields single crystals of a 2D network of formula {[K_2_(dmso)(H_2_O)_5_][Ni_2_(H_2_mpba)_3_]∙dmso∙2H_2_O}*_n_* (**1**), whereas in the case of a H_2_O/MeOH (1:1 *v*/*v*) mixture, another compound of formula [Ni(H_2_O)_6_][Ni_2_(H_2_mpba)_3_].3CH_3_OH.4H_2_O (**2**) results. Finally, we sought to assess the effects of the counter cations on the electronic properties of these molecular materials using XANES spectroscopy, electrochemical measurements in the solid state, and theoretical DFT-type calculations.

## 2. Results and Discussion

### 2.1. Synthesis, IR Spectroscopy, Thermal Analysis, and X-ray Powder Diffraction

Both compounds were prepared by the diffusion method. The heterometallic K^I^/Ni^II^ coordination polymer **1** was obtained from the reaction between NiCl_2_∙6H_2_O and K_2_H_2_mbpa in a H_2_O/MeOH/dmso solvent mixture. Its structure is made up of triple-stranded dinickel(II) units that are interlinked by centrosymmetric dipotassium(I) entities. The same reaction in a binary mixture of polar protic solvents such as methanol and water yielded compound **2** where the negative charge of the discrete triple-stranded dinickel(II) anionic complex is counter-balanced by the hexaaquanickel(II) complex cation. These results point out that the use of polar aprotic solvents like dmso (this work) or dimethylformamide (dmf) as in a previous report where the compound of formula {[K_2_(dmf)_2_(H_2_O)_2_][Ni_2_(H_2_mpba)_3_]∙2H_2_O}*_n_*, was isolated) [21] appears to be responsible for the self-assembly of the [Ni_2_(H_2_mpba)_3_]^2−^ motif in a heterometallic framework (see Figure 1).

The thermal analysis of **1** (Appendix A) shows weight losses corresponding to three endothermic processes at 66, 106, and 276 °C which are attributed to the release of seven water molecules (two being free and the other five coordinated) and two dmso molecules in the temperature range of 25–291 °C (obsd. 24.5%; calcd. 23.5%). Moreover, other mass losses occur in the range of 291–505 °C which are due to the thermal decomposition of 2.5 H_2_mpba^2‒^ ligands with exothermic peaks in the DTA curve centered at 360, 398, and 472 °C (obsd. 52.4%; calcd. 50.9%). The stable residue at 498 °C (22.4%) is most likely related to a mixture of 2NiO and K_2_O (19.8%) with an organic material, which is difficult to identify. The thermogravimetric study of **2** (Appendix A) shows a first-mass loss in the temperature range 89–110 °C, which is accompanied by an endothermic event in the DTA curve at 90 °C. This feature can be attributed to the removal of uncoordinated solvent molecules (obsd. 14.2; calcd. 14.0%). This mass loss continues in the TG curve until 502 °C due to the release of coordinated water molecules and thermal decomposition of the organic ligand. The DTA curve exhibits two exothermic processes at 393 and 456 °C. An amount of residue can be noted at 840 °C which is most likely due to NiO plus the organic moiety.

The FT-IR spectra of **1** and **2** show characteristic absorption bands centered at 3340 and 3240 cm^–1^ (**1**) and 3300 cm^–1^ (**2**) which are assigned to the amide ν(NH) stretching vibrations suggesting the occurrence of the partial deprotonated H_2_mpba^2−^ form as ligand. The strong absorption peaks at 1632 and 1603 cm^–1^ (**1**) and 1630 and 1600 cm^–1^ (**2**) were attributed to ν(CO) stretching vibrations (to be compared with the peaks at 1677 and 1609 cm^–1^ occurring in the i.r. spectrum of the K_2_H_2_mpba salt). These assignments are, in turn, associated with the carbonyl-amide and carboxylate groups from the H_2_mpba^2−^ ligand. Moreover, bands around 3400 cm^−1^ relative to ν(OH) from water molecules were observed in the i.r. spectra of **1** and **2**. Finally, the absorption peaks at 1011 and 939 cm^–1^ which are observed in the spectrum of **1** and that are lacking in **2** are attributed to the ν(SO) stretching of uncoordinated dmso molecules [34].

To confirm the phase purity of the bulk materials, X-ray powder diffraction (XRPD) experiments were carried out for **1** and **2** (Appendix A), and the good coincidence in the position of all peaks confirmed that the single crystal X-ray structures are the same as the as-synthesized bulk materials **1** and **2**.

### 2.2. Description of the Crystal Structures of ***1*** and ***2***

The crystal structures of **1** and **2** were determined by single-crystal X-ray diffraction, as provided below. Compound **1** crystallizes in the monoclinic crystal system in the *P*2_1_*/n* space group while **2** crystallizes in the orthorhombic crystal system in the *Fddd* space group. Crystal data and details of the data collection and refinement for **1** and **2** are listed in Table 1. Selected bond lengths and angles plus hydrogen bonds and C-H^…^O type contacts are given in Appendix A. SHAPE software calculations [35] indicated that the coordination geometry of all the Ni^II^ atoms in 1 and 2 is a distorted octahedron (Oh) with different distortion degrees. Additionally, the coordination environment for K1 in **1** is Snub disphenoid J84 (D_2d_) and the other K2 is a distorted octahedron (O_h_). The results through the continuous shape measures (CShMs) can be seen in Appendix A.

A perspicuous representation of the coordination environment of **1** (Figure 2) shows how its H_2_mpba^2−^ ligands exhibit three different coordination modes with the two crystallographically independent Ni^II^ centers (Ni1 and Ni2) being separated by a distance of6.302 (1) Å. The nickel (II) environments exhibit the same chirality as the [Ni_2_(H_2_mpba)_3_]^2−^ units, ΔΔ or ∧∧ (the centric crystal structure leadings to a mixture of both enantiomers in the single crystal), resulting in an helicate building block [36,37,38,39,40,41,42]. The two oxamidate groups of each H_2_mpba^2−^ ligand exhibit a *syn*-*syn* configuration around the central phenylene ring. The three coordination modes of the H_2_mpba^2−^ligands involve different chemical environments leading to a 2D-coordination network which expands along the diagonal of the crystallographic *ac* plane, where each helicate unit is connected to two different types of dinuclear units based on solvated K^+^ ions. The H_2_mpba^2−^ ligand in **1** is versatile, as shown by the different bridging modes between the Ni^II^/K^I^ ions (see Appendix A for further details). For example, O11 and O17, among other sets of atoms not highlighted herein, form oxo-bridges connecting the Ni2/K1^iv^ and Ni1/K2^iii^ pairs, respectively.

In this way, the triple-stranded [Ni_2_(H_2_mpba)_3_]^2−^ anionic motif in the polymeric structure of **1** is stabilized by hydrogen bonds involving uncoordinated water and dmso molecules and coordinated dmso, exhibiting a motif in which the dinickel(II) units are connected to five neighboring K^I^ ions (Appendix A). Thus, a 2D coordination network grows along the diagonal of the crystallographic *ac* plane connecting each helicate unit to two different types of dinuclear units. Appendix A shows each generic helicate **M** surrounded by the other six ones (noted as **M1**–**M6**) with the values of the shortest and largest Ni···Ni inter-helicate distances equal to 8.111(1) Å and 8.660(1) Å, respectively [21]. To the best of our knowledge, the spatial arrangement of the K^I^-dinuclear units in the network admits a regular and periodic array which is represented by an underlying net with **sql** topology [43,44] (Appendix A). The simplification method in the standard mode establishes centroids on the K^I^-dinuclear units that are 4-connected nodes (Appendix A).

In the crystal structure of **2**, three amidate-oxygen atoms (O2/O4/O8) plus three carboxylate-oxygens (O1/O5/O7) from three oxamate groups are coordinated to Ni1 and the same occurs at Ni1^ii^, the donors being the symmetry-related O2^ii^/O4^ii^/O8^ii^ and O1^ii^/O5^ii^/O7^ii^ sets of oxamate-oxygen atoms (Figure 3). The Ni1 and Ni1^ii^ atoms are six coordinated by three distinct deprotonated H_2_mpba^2−^ ligands with the same coordination mode, each metal ion exhibiting a racemic crystal with ΔΔ or ΛΛ conformers. The Ni1···Ni^ii^ distance observed in **2** is 6.3121(7) Å, a value which is *quasi*-identical to that observed in **1** [ca. 6.30 Å], this feature suggesting that the interaction of the [Ni_2_(H_2_mpba)_3_]^2−^ helicate with the K^+^ ions in **1** does not cause any significant intra-helicate elongation. The negative charge of the [Ni_2_(H_2_mpba)_3_]^2−^ triple-stranded unit in **2** is balanced by the [Ni(H_2_O)_6_]^2+^ counter ion where the metal center exhibits a slightly distorted octahedral surrounding (see Figure 3 and Appendix A).

The supramolecular assessment of hydrogen bonds and contacts was carried out based on geometric parameters (see Appendix A). The shortest inter-helicate Ni···Ni distance in **2** is 9.759(1) Å and the largest one is 13.8696(8) Å, both values being greater than those observed in **1** [8.111(1) and 8.660(1) Å, respectively]. A GraphSet R_2_^2^(10) observed in these patterns involves the N1-H1···O3^iii^ and N2-H2···O6^iv^ hydrogen bonds. The supramolecular 3D arrangement is built through the [Ni(H_2_O)_6_]^2+^ cationic complex, which acts as a hydrogen bond donor via O10-H10B···O5, O11-H11B···O7, and O12-H12B···O1 interactions displaying the R_2_^2^ (8) GraphSet (see Figure 3).

### 2.3. Theoretical Study

The frontier orbitals and their corresponding calculated energies are shown in Appendix A. The results in the case of **2** show two unpaired electrons for each Ni^II^ ion, with each pair lying on nearly degenerate molecular orbitals that are mainly made up of the 3*d* orbitals of Ni^II^, as expected from the classical molecular orbital theory. However, the calculations predicted a very different picture for **1**, with the singly occupied HOMO-diffusely spread on the planar ligand having a π character. This possibly accounts for the presence of the pre-edge feature at the XANES spectrum of **1**, which indicates more hybridization of 3*d*–4*p* orbitals, and thus the deviation from the simple *d*-type (Appendix A). Furthermore, the absence of this pre-edge feature in **2** agrees with the conventional *d* character of all singly occupied orbitals in this compound, which also leaves the 4*p* orbitals of **2** free for the more intense dipolar peak, as observed. The intriguingly-shaped HOMO of **1** is substantially more weakly bonding than that of **2** (see ESI). The HOMO of **1** is non-degenerated and it spreads around the ligand instead of on the metal center showing a π character. The HOMO-1 and HOMO-2 are also singly occupied but can be largely described as 3d orbitals from the Ni atom. As for **2**, all singly occupied molecular orbitals (SOMOs) are mainly made by nickel 3d orbitals. For this reason, the HOMO orbital of **2** is more tightly bound (−3.21 eV to be compared with −2.8 eV for **1**). We have also calculated the vertical ionization energy, which corroborates this trend: the energy required to remove an electron from **1** is 6.26 eV while it amounts to 7.72 eV for **2**, corroborating the fact that it is much easier to remove one electron from the diffuse π orbital of **1**.

### 2.4. Electrochemical Behavior of the Supramolecular Materials

In this section, the electrochemical behavior of the supramolecular materials is discussed. Figure 4 presents the cyclic voltammograms for **1** (A) and **2** (B) obtained at different scan rates.

Both complexes are redox-active and exhibit faradaic signals referring to the Ni^I^/Ni^II^ pair, but the structural and electronic differences affect the polarization and profile of the CV curves. The charge transfer for **2** occurs to a greater extent and the highest current densities can be correlated to the greater number of nickel(II) ions per formula. In this sense, the electrochemical processes involve the redox centers of the helicate and the complex cation in **2**. Furthermore, the ratio between the intensities of the anodic and cathodic peaks is closer to unity for **2** (I_anodic_/I_cathodic_ = 1.46, at 5 mV s^–1^) compared to **1** (I_anodic_/I_cathodic_ = 1.94, at 5 mV s^–1^), indicating a greater electrochemical reversibility.

Another important difference between **1** and **2** concerns the values of their formal potentiasl (*E*_1/2_). They are +0.42 and +0.37 V (vs. Ag/AgCl/3.5 M KCl) for **1** and **2**, respectively. Changes in the formal potentials reflect the energy changes in the molecular orbitals involved with the electron transfer, corroborating the results predicted by the DFT calculations discussed above. The voltammetric measurements also reveal that the electrochemical reactions on the electrodes have diffusional control for both complexes. This is attested by the linear dependence of the peak current (I_p_) upon the square root of the scan rate, ν^1/2^ (Figure 4c,d). Based on this behavior, it can be inferred that the OH^−^ anions play a crucial role in the reactions, justifying the diffusional control (mass transport), interacting and also maintaining the electroneutrality of the materials after the change in the oxidation state. To support this hypothesis, voltammetric measurements were also conducted in neutral KCl electrolytes (Appendix A). CV curves in this medium do not show any signal of the Ni^I^/Ni^II^ pair and they are mostly capacitive. This demonstrates that the chloride anions, being bulkier and having a lower charge density, are not able to interact with the complexes and stabilize them.

Considering the participation of OH^−^ species in the redox reactions, there are two possibilities: (i) being a strong field ligand, OH^−^ ions can replace one of the coordinated oxygen atoms at the coordination sphere of the Ni^II^ ions in **1** and **2**, and also replace the water molecules in the complex cation of **2**; and/or (ii) electrostatic interactions can occur between OH^−^ ions and low-electron density sites in the supramolecular structure.

### 2.5. XANES Spectroscopy Study

We have conducted a further study to shed light on the electronic structure of **1** and **2**. X-ray absorption near-edge structure spectroscopy (XANES) was used to investigate the structures around the Ni K-edge [Figure 5a]. For this edge, a 1s core electron is excited to an available state above the HOMO (4p level) by an X-ray photon in a dipolar approximation. However, this electron can also be promoted to a Ni 3d prohibited final state, corresponding to the quadrupolar contribution. Comparing the spectra of **1** and **2**, one can notice that the dipolar peak is more significant for **2** [see Figure 5b], indicating the higher contribution of the Ni-4p empty states for this compound. This could suggest that the Ni-4p states for this sample are more localized than those in **1**. The quadrupolar transition can occur under peculiar conditions, namely, when the hybridization between the 4p-nd levels is likely. In a qualitative analysis, the pre-edge feature observed on **1**, located just a few eVs below the edge [around 8.328 keV; see Figure 5c], can be associated with a quadrupolar transition (from 1s to a nd). This pre-edge feature in the spectrum of **1** indicates that the 3d orbitals from nickel (II) are hybridized with the np levels. In contrast, the main peak, the Ni K-edge, is related to the dipolar one (1s to a np), as shown in Figure 5.

Thus, XANES performed on the Ni K edge of both compounds suggest a scenario where depending on the Ni^II^ complex environment, the system can present more localized Ni-4p or more spread 3d Ni states, which can give rise to a 4p-nd hybridization. The enhancement of the white line observed for **2**, can signal the location of the 4p states. On the other hand, a 4p-nd hybridization is attributed to the observed pre-edge feature for **1**.

## 3. Materials and Methods

### 3.1. Reagents

Nickel (II) chloride hexahydrate, potassium hydroxide, dimethyl sulfoxide, acetone, ethanol, and methanol were purchased from commercial sources and used without further purification. The Et_2_H_2_mpba proligand was prepared as previously described [21,25].

### 3.2. Preparation of the Oxamate Ligand and Complexes ***1*** and ***2***

K_2_H_2_mpba. This salt was prepared by a previously reported procedure [21]. Briefly, an aqueous solution (2.0 mL) of KOH (0.180 g, 3.2 mmol) was added to an aqueous suspension (12.0 mL) of Et_2_H_2_mpba (0.5 g, 1.6 mmol) at 60 °C under continuous stirring during for 30 min. The resulting clear solution after the hydrolysis reaction was cooled to 25 °C. The addition of 15 mL of acetone caused the precipitation of a white solid which was filtered off on a sintered glass filter and air-dried. Yield: 85%. IR ( cm^–1^): 3358 [ν(_N‒H_)], 1677; 1609 [ν(_C=O_) _+_ ν_as_(COO^–^)], 1542; 1510 [ν(_C‒C_)], 1484 [ν_s_(COO^–^)], 1386; 1373 [ν(_C‒N_)] and 878; 786 [ν(_C‒H_)].

{[K_2_(dmso)(H_2_O)_5_][Ni_2_(H_2_mpba)_3_]∙dmso∙2H_2_O}*_n_* **(1)**. A dmso solution (2.0 mL) of NiCl_2_∙6H_2_O (0.022 g, 0.091 mmol) was placed at the bottom of a test tube. Then, a layer of methanol (2 mL) was added followed by an aqueous solution (2.0 mL) of K_2_H_2_mpba (0.030 g, 0.091 mmol) which was carefully layered on the top. The tube was covered with parafilm^®^ and left to diffuse at room temperature. Needle-like green X-ray quality crystals of **1** were collected by filtration after 6 days. Yield: 40%. Anal. Calcd for C_34_H_44_K_2_N_6_Ni_2_O_27_S_2_ (**1**): C, 31.5; H, 3.6; N, 6.8. Found: C, 32.3; H, 3.4; N, 6.7%. IR (cm^‒1^): 3414 [ν(_O‒H_)], 3340; 3240 [ν(_N‒H_)], 3187, 3075; 2912 [ν(_C‒H_)], 1666; 1610 [ν(_C=O_) _+_ ν_as_(_COO_)], 1592; 1555 [ν(_C=C_)], 1458; 1355 [ν_s_(_COO_)], 1289 [ν(_C‒N_)], 1011; 939 [ν(_S=O_)], and 689 [δ(_C‒H_)].

[Ni(H_2_O)_6_][Ni_2_(H_2_mpba)_3_]∙3CH_3_OH∙4H_2_O **(2)**. Crystals of **2** were obtained by slow diffusion of the reagents in a test tube as done for **1**. An aqueous solution (3.0 mL) of K_2_H_2_mpba (0.030 g, 0.091 mmol) was deposited at the bottom of a test tube and a layer of a water/methanol mixture (2.0 mL, 1:1 *v*/*v*) was placed on the top. Then, a methanolic solution of NiCl_2_∙6H_2_O (0.022 g, 0.091 mmol) was carefully added forming the third layer. The tube was covered with parafilm^®^ and allowed to diffuse at room temperature. Block-shaped green crystals of **2** suitable for X-ray diffraction were collected after 25 days. Yield: 30%. Anal. Calcd for C_33_H_50_N_6_Ni_3_O_31_ (**2**): C, 33.7; H, 4.19; N, 6.99. Found: C, 34.4; H, 3.93; N, 7.07%. IR ( cm^‒1^): 3429 [ν(_O‒H_)], 3300 [ν(_N‒H_)], 3148, 3086; 2962 [ν(_C‒H_)], 1620, [ν(_C=O_) _+_ ν_as_(COO)], 1553; 1475 [ν(_C=C_)], 1458; 1369 [ν_s_(_COO_)], 1289 [ν(_C‒N_)], and 684 [δ(_C‒H_)].

### 3.3. X-ray Data Collection and Structure Refinement

Single crystals of **1** and **2** were collected at 293(2) K on a Bruker-AXS Kappa Duo diffractometer with an APEX II CCD detector (**1**) and on a Bruker D8 Venture diffractometer (**2**). The absorption correction for **1** was carried out with an analytical numeric absorption correction using a multifaceted crystal model [45]. APEX3 [46] and SADABS [47] were employed for the data collection, cell refinement, data reduction, and multi-scan method-absorption correction in the case of **2**. The solution and full–matrix least–squares refinements based on *F*^2^ were performed with the SHELXS [48] and SHELXL [45] programs, respectively, included in the WinGX [49] (**1**) or Olex2 (**2**) software [50]. All non-hydrogen atoms were identified and refined by the least-squares full-matrix on *F*^2^ with anisotropic thermal parameters. Hydrogen atoms were included as fixed contributions according to the riding model. The MERCURY program [51] was used to prepare artwork representations. The topological analyses of the networks were performed by the ToposPro package [52,53] at the standard method for the simplification process. Crystals of **2** are air-sensitive when removed from the mother liquor after 15 days. Besides, the crystal structure of **2** displays large accessible voids, which are filled by disordered solvent molecules of crystallization.. All our attempts to unequivocally identify them failed and the solvent mask routine implemented on the Olex2 program was used [50]. A total of 1504 electrons were found in a volume of 6920 Å^3^ at one void per unit cell. This is consistent with the presence of three methanol and four water molecules per formula unit which would account for 1504 electrons per unit cell.

### 3.4. Physical Measurements

IR spectra were recorded on a Thermo Scientific iS50 (Waltham, MA, USA) spectrophotometer coupled to Pike Gladi FTIR in the wavenumber range 4000–400 cm^−1^ with an average of 144 scans and 4 cm^−1^ of spectral resolution using an ATR apparatus. Elemental analyses were carried out with a CHNS/O Elemental Analyzer Perkin Elmer 2400 (Waltham, MA, USA). Thermal analyses (TGA/DTA) were performed with the same modulus employing a thermobalance Hitachi (EXSTAR SII TG/DTA 7300 Tokyo, Japan) in the temperature range 35–1100 °C by using alumina crucibles and around 3 mg of each sample. A dinitrogen flow of 100 mL min^‒1^ with a heating rate of 10 °C min^‒1^ was used. Powder X-ray diffraction data for **1** and **2** were collected at room temperature using a Rigaku Ultima IV with Cu-Kα radiation (λ = 1.5418 Å at 40 kV and 30 mA) and 2θ from 3 to 50°, the step size being 0.02° (Appendix A).

### 3.5. Theoretical Calculations

Density functional theory calculations were performed with the *ω*B97X-D functional with the SNKJC basics set including an effective core potential. All calculations were carried out using the GAMESS-US package and employed restricted wavefunctions, thus avoiding spin contamination. Several spin multiplicities were tested to determine the ground state [54,55,56]. Starting from the structures obtained by the X-ray diffraction, we extracted the smallest molecule size able to mimic the complex and its counterions. The calculations were performed on an isolated system, without any charge. Thus, when simulating **1**, we have included two K^I^ cations to balance the doubly-charged negative complex anion of the complex, whereas the simulation of **2** included a single [Ni(H_2_O)_6_]^2+^ counterion. The potassium cations from the CIF file were bound to more than one H_2_mpba^2−^ entity, and since our model system consists of only one of such units, the oxygen atoms participating in the K-O bonds with the vicinal H_2_mpba^2‒^ ligands were passivated as K-OH_2_. The energy of singlet, triplet, quintet, and septet spin multiplicities were calculated to identify the preferred spin state of each molecule. As a result, **1** was found to display a quintet ground state (four unpaired electrons) while **2** displayed a septet one (two unpaired electrons per Ni atom, or six unpaired electrons overall). Geometry optimizations were performed in the ground states of **1** and **2** to relax the initial structure extracted from the X-ray diffraction data.

### 3.6. Electrochemical Experiment Set Up

The working electrodes were prepared from slurries composed of the supramolecular materials (**1** and **2**), MWCNT (Nanocyl NC 7000), and polyvinylidene fluoride (PVDF) in an 8:1:1 mass ratio, using 1-methyl-2-pyrrolidone (NMP) as a solvent. Due to the low electrical conductivity of the active complexes, multi-walled carbon nanotubes (MWCNT) were used as a conductive additive and PVDF as a binder. The mixture was stirred for 12 h (350 rpm) and subjected to ultrasonic treatment for 15 min. Finally, 100 µL of the slurry was added dropwise over the current collector (gold disks with 2.26 cm^2^) followed by drying for 12 h at 85 °C on a hot plate to allow the complete evaporation of NMP. The electrodes presented masses of approximately 1.5 mg. Cyclic voltammetry (CV) experiments were conducted using an Omnimetra PG 29A Potentiostat (National Instruments software, Austin, TX, USA) at 25 °C in three-electrode configuration cells. 1.0 M KOH was used as electrolyte, graphite as the counter electrode, and Ag|AgCl|KCl (3.5 M) as the reference electrode.

### 3.7. XANES Study

The XANES measurements were conducted at the Extreme Methods of Analysis (EMA) beamline [57], sited at 4th generation Brazilian synchrotron Sirius. The X-ray EMA source is a 22 mm period Kyma undulator which delivers photons in the energy range from 5 (3rd harmonic) to 30 keV (13th harmonic), selected by the suitable choice undulator phase. The outgoing beam is monochromatized by an N2L-cooled high-resolution Double Crystal Monochromator (DCM). The EMA’s DCM has two sets of silicon crystals ([111] or [311]). Finally, an achromatic set of K-B mirrors focuses the beam at the sample position with a spot size down to 1.0 × 0.5 um. This beamline is in the final stage of commissioning. It is equipped with the complete infrastructure to study the crystalline, magnetic, and electronic structure of materials under extreme temperature, magnetic field, and pressure conditions. The K-edge of the Ni of both samples was measured at the undulator 3rd harmonic using the [111] crystals of the DCM with a spot size of 10 × 10 µm at the sample. The spectra were collected in a transmission configuration. The intensity of the incident beam (*I*_0_) was measured using a semitransparent photodiode and the transmitted one (*I*) through a thicker photodiode with good sensitivity in this range of energy. The absorption spectra were calculated using a Beer-Lambert law [ABS = *ut* = ln(*I_0_*/*I*), where *u* is the absorption coefficient of the sample and *t* is its thickness]. Ten spectra were collected for each sample, each one of these spectra being individually normalized by the absorption jump and averaged to obtain the result.

## 4. Conclusions

In summary, we show here a simple synthetic protocol at room temperature which is suitable for the preparation of two new crystalline supramolecular materials bearing the triple-stranded [Ni_2_(H_2_mpba)_3_]^2−^ helicate, {Ni^II^_2_}. The presence of either K^+^ (**1**) or [Ni(H_2_O)_6_]^2+^ (**2**) as counter cation in the supramolecular structures was likely driven by the different solvent mixtures used in the slow diffusion systems containing the same starting materials (K_2_H_2_mpba and NiCl_2_∙6H_2_O). The coordination mode adopted by each H_2_mpba^2−^ ligand around the Ni^II^ ions in **1** and **2** involves only oxygen atoms from the two monoprotonated oxamate moieties forming the [Ni_2_(H_2_mpba)_3_]^2−^ dianionic helicate without deprotonation of the amide-nitrogen atoms. This feature allows for the establishment of hydrogen bonds, which play a key role in the crystal packing of both compounds. The {Ni^II^_2_} helicate is connected by K^+^ counter cations in the crystal structure of **1** affording a 2D coordination network with a **sql** topology, and in contrast to **1**, three neighboring {Ni^II^_2_} helicate units in **2** interact in a supramolecular fashion through four homo *R*_2_^2^(10) synthons yielding a supramolecular 2D array. The structural and electronic differences between **1** and **2** are reflected in the electrochemical response. The Ni^II^ ions from the helicate and the counterion (complex cation) in **2** can be reversibly reduced, resulting in the highest faradaic current intensities. The redox reactions in **1** also occur in an alkaline medium but at higher formal potentials. For **2**, the charge transfer occurs to a greater extent and the highest current densities can be correlated to the highest number of nickel(II) ions per formula. Changes in the formal potentials reflect energy changes in the molecular orbitals that are involved with the electron transfer, and this corroborates the results predicted by the theoretical DFT-type calculations. XANES results suggest a scenario quite susceptible to the Ni^II^ complex environment. For **1**, the XANES signal can be associated with a localized Ni-4p. On the other hand, the results for **2** indicate a spread of 3d Ni states which gives rise to a 4p-nd hybridization.

We hope our findings will open an avenue to the fine-tuning of the redox properties of coordination compounds and will contribute to the discovery of new redox-active molecular materials being triggered by counterions.

## Data Availability

Not applicable.

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
