# Peer review of "Study of the Counter Cation Effects on the Supramolecular Structure and Electronic Properties of a Dianionic Oxamate-Based {NiII2} Helicate"

_molecules, 2023, doi:10.3390/molecules28052086_

Round 1

Reviewer 1 Report

The manuscript entitled “Study of the Counter-Cation Effects on the Supramolecular Structure and Electronic Properties of a Dianionic Oxamate-Based {NiII2} Helicate” by Julve and Marinho et al described the synthesis, crystal structure, and electronic properties of {[K2(dmso)(H2O)5][Ni2(H2mpba)3]·dmso·2H2O}n and [Ni(H2O)6][Ni2(H2mpba)3]·3CH3OH·4H2O bearing the [Ni2(H2mpba)3]2- helicate. Due to the counter-cation effects, these two compounds exhibited different structures and properties. This manuscript is helpful in understanding the influence of the counter cations. Some minor issues as commented below have to be addressed before it can be accepted for publication.

1. The "sq1 topology" in the Abstract section should be a typo, please correct it to "sql topology".
2. One of the benzene rings of compound 1 in Scheme 1 does not have double bonds drawn.
3. Abbreviations such as mpba, dmso, dmf and so on should be given in full the first time they are used, either in the Text or in the Abstract.
4. The font sizes of the text in the figures of this manuscript are all on the small side and it is recommended that they should be adjusted.
5. Figure 4 is missing the (A)(B) mark.
6. It is recommended that the inset images for Figures 4 and 5 should be split out.
7. References related to constructions of helical coordination compounds should be cited, such as CrystEngComm, 2010,12, 324. Similarly to the counter ions, examples for solvent effects are worth discussing (Chem. Commun., 2005, 2232; Chin. J. Chem. 2021, 39, 2718.).

Author Response

Dear Prof. Xie,

Thank you very much for your e-mail on February 16th, 2023 informing us that minor revisions are necessary before acceptance of our manuscript in Molecules.

We are submitting the revised version of our manuscript, and we are glad to inform you that all of the changes/comments/suggestions raised by the reviewers have been taken into account. All our additions and changes are highlighted in yellow in both manuscript and ESI for an easy assessment. A cover letter with the changes made in the revised version was upload as an attached file.

 Best regards

Prof. Maria Vanda Marinho

Reviewer 2 Report

The manuscript by Julve, Marinho, and coworkers focuses on the synthesis, characterization, and theoretical studies of two Ni(II) complexes, 1 and 2, featuring relevant electronic and structural properties related to their supramolecular organization, with particular emphasis on the role of the counter-cation associated to the {NiII2}2- helicate.

Exhaustive descriptions of the X-ray structures of both compounds are provided, together with the discussion of IR and XANES spectra, thermal analyses, and cyclic voltammetry. Theoretical calculations have also been performed to identify the frontier molecular orbitals of 1 and 2.

The manuscript is well written and the results are clearly presented, with the exception of some minor points that are commented below. Nevertheless, I believe that the content of this valuable piece of research would be mostly of interest to a readership specifically dedicated to coordination chemistry. Therefore, my overall suggestion is to consider publication of the manuscript in a more specialized journal, such as Inorganics.

Further suggestions:

-      Improve Figure 1 and Scheme 1 (font size, stereo-bonds, etc.)

-      Line 137: remove “and”

-      Line 143: “with an organic material”

-      Line 153: “stretching vibrations”

-      Avoid indicating the three non-equivalent structures of the ligand found in 1 as L1-L3. It might be misleading.  

-      Line 289: “The intriguing… more weakly bound than …”. I assume that the authors use “bound” as “bonding”, which sounds more correct.

Author Response

(The authors gave the same response as above.)
